# A Novel Monopole Ultra-Wide-Band Multiple-Input Multiple-Output Antenna with Triple-Notched Characteristics for Enhanced Wireless Communication and Portable Systems

**DOI:** 10.3390/s23156985

**Published:** 2023-08-06

**Authors:** Shahid Basir, Ubaid Ur Rahman Qureshi, Fazal Subhan, Muhammad Asghar Khan, Syed Agha Hassnain Mohsan, Yazeed Yasin Ghadi, Khmaies Ouahada, Habib Hamam, Fazal Noor

**Affiliations:** 1School of Engineering & Applied Sciences, ISRA University, Islamabad 44000, Pakistan; shahid.basir@yahoo.com; 2School of Optics and Photonics, Beijing Institute of Technology, Beijing 100081, China; ubaid_ur_rahman@outlook.com; 3Department of Technology, Abasyn University Islamabad Campus, Islamabad 44000, Pakistan; fazal.subhan@abasynisb.edu.pk; 4Faculty of Engineering Sciences and Technology, Hamdard University, Islamabad 44000, Pakistan; m.asghar@hamdard.edu.pk; 5Optical Communications Laboratory, Ocean College, Zhejiang University, Zheda Road 1, Zhoushan 316021, China; 6Department of Computer Science, Al Ain University, Al Ain 64141, United Arab Emirates; yazeed.ghadi@aau.ac.ae; 7Department of Electrical and Electronic Engineer Science, School of Electrical Engineering, University of Johannesburg, Johannesburg 2006, South Africa; kouahada@uj.ac.za; 8College of Computer Science and Engineering, University of Hail, Hail 55476, Saudi Arabia; 9Faculty of Engineering, Université de Moncton, Moncton, NB E1A 3E9, Canada; 10International Institute of Technology and Management (IITG), Avenue des Grandes Ecoles, Libreville BP 1989, Gabon; 11Faculty of Computer and Information Systems, Islamic University of Madinah, Madinah 400411, Saudi Arabia; mfnoor@iu.edu.sa

**Keywords:** UWB, MIMO, CSRR, C-shaped, MIMO diversity, high isolation, triple-notched band

## Abstract

This study introduces a monopole 4 × 4 Ultra-Wide-Band (UWB) Multiple-Input Multiple-Output (MIMO) antenna system with a novel structure and outstanding performance. The proposed design has triple-notched characteristics due to CSRR etching and a C-shaped curve. The notching occurs in 4.5 GHz, 5.5 GHz, and 8.8 GHz frequencies in the C-band, WLAN band, and satellite network, respectively. Complementary Split-Ring Resonators (CSRR) are etched at the feed line and ground plane, and a C-shaped curve is used to reduce interference between the ultra-wide band and narrowband. The mutual coupling of CSRR enables the MIMO architecture to achieve high isolation and polarisation diversity. With prototype dimensions of (60.4 × 60.4) mm^2^, the proposed antenna design is small. The simulated and measured results show good agreement, indicating the effectiveness of the UWB-MIMO antenna for wireless communication and portable systems.

## 1. Introduction

The demand for high data rates in reliable communication has driven advancements in RF and microwave engineering, particularly in the field of wireless communication, which is widely acknowledged as one of the most significant contributions to the technology world. This technological revolution has resulted in the widespread use of digital data. To meet these increasing expectations, Ultra-Wide-Band (UWB) technology was introduced, offering improved communication accuracy, speed, and efficiency of data transmissions. The appeal of UWB technology has grown over the past few decades due to its desirable features such as accuracy, high data rates, and cost-effectiveness. The Federal Communications Commission (FCC) defines UWB within the frequency range of 3.1–10.6 GHz [1,2]. UWB technology finds applications in various wireless communication domains, including radar systems, medical applications, and personal communications [3,4,5]. The enhancement of antenna characteristics such as antenna gain, bandwidth, and radiation pattern has made UWB particularly suitable for Wireless Capsule Endoscopy (WCE) systems [6,7]. Interference issues arise when UWB signals coexist with narrow bands like WiMax, C-bands, and WLAN. Notching techniques have been employed to mitigate interference between these frequency bands. Previous studies have proposed compact UWB trapezoidal antennas with dual-band notching characteristics [8,9,10,11] and UWB antennas with triple notching [12,13,14,15,16] to address this issue. These techniques involve the insertion of comb slot shapes into the radiating patch to achieve notching, while the UWB characteristics are achieved through ground structure modifications in a staircase form [17,18,19,20]. In addition, UWB uses a variety of techniques for notching [21,22,23,24,25,26], and switches and stubs are embedded in the slot antenna [27,28,29,30,31]. Moreover, MIMO (Multiple-Input Multiple-Output) technology was developed to enhance data transmission efficiency, communication quality, and capacity. However, it introduced coupling effects between antennas. To overcome this challenge and improve the quality of RF signals while utilizing available bandwidth efficiently, different techniques have been introduced in MIMO systems. For instance, CSRR (Complementary Split-Ring Resonator) slots and J-shaped slots have been utilized for notching dual bands [32], while coupling reduction has been achieved through resonating slots on the ground or isolation elements placed between resonating elements [33,34]. In a co and cross configuration, periodic S-shaped DGS units (PSDGS) have been placed between square patch elements to induce high isolation [35]. Dielectric Resonator Antenna (DRA) techniques, Complementary Split-Ring Resonators (CSRR), neutralization techniques, and stub and T-slot integration into the ground have been proposed to block surface current flow and improve isolation [36,37,38,39]. The use of a varactor diode enables capacitive tuning for reconfigurable antennas. Various techniques such as varactor diode and biasing circuit, Electromagnetic Band Gap (EBG), and Substrate-Integrated Backed Cavity Slots (SIBCS) have been proposed to reduce mutual coupling in MIMO systems [40,41,42,43,44,45].

Despite these advancements, challenges remain in terms of antenna size, multiple notching, and additional elements for coupling reduction. In this paper, we present a 4 × 4 crescent-shaped monopole UWB-MIMO antenna system that addresses these size and notching challenges and reduces coupling without the need for additional elements. The proposed system achieves multiple notches for C-band, WLAN, and satellite networks through the etching of Complementary Split-Ring Resonators on the patch elements, ground side, and C-shaped structures. The MIMO antenna system is arranged in an orthogonal configuration to achieve pattern diversity and polarization diversity. The proposed design covers a bandwidth from 2 GHz to 13 GHz. The notching characteristic enables the mitigation of coexisting sub-bands in the UWB region, reducing interference without the complexity of additional circuits. Moreover, the proposed design achieves high isolation in the MIMO system through the mutual coupling of Complementary Split-Ring Resonators, resulting in a miniaturized design. The multitasking of CSRR makes the article novel and differentiable. This miniaturized design is particularly suitable for wireless communication. The multitasking of CSRR is organized in the manuscript as follows: In Section 1, an introduction to UWB, notching characteristics, and the MIMO antenna system is presented; Section 2 is about the development and design of the antenna, and the result of the UWB, notching, and MIMO system are presented; Section 3 presents a comparative and theoretical analysis of the proposed antenna with other designs on the basis of notching characteristics and isolation; and Section 4 provides the conclusion of the article.

## 2. Proposed Antenna Design

A monopole UWB-MIMO antenna system with triple notching features for C-band, WLAN, and satellite networks is presented. The notching characteristic is achieved by etching Complementary Split-Ring Resonators (CSRR) at the patch elements and ground side, and the third notch is achieved with the C-shape. The 4 × 4 MIMO antenna system is arranged orthogonally to achieve pattern and polarisation diversity. Because of the symmetric positioning of the antenna port components, as well as the mutual coupling of Complementary Split-Ring Resonators, which block the RF signal between the port elements, the MIMO antenna obtained good isolation. In this article, the etching of CSRR achieves both notching qualities and excellent isolation in the MIMO system. The proposed antenna is fabricated on an FR-4 substrate with a dielectric constant of 4.4, loss tangent of 0.02, and a thickness of 1.6 mm. The geometry and dimensions of the monopole, 4 × 4 MIMO, CSRR, and Arch shape or C-shape are illustrated in Figure 1. The antenna design is modeled and simulated using ANSYS HFSS-21. The physical prototype of the proposed antenna is depicted in Figure 1d. The leading edge of the patch is crescent-shaped and incorporates a CSRR in the feeding line, while another CSRR is embedded in the ground plane. The research work is carried out in the following steps:The UWB design achieves a bandwidth of 2 GHz to 13 GHz by feeding a rectangular patch with a ground plane.Notching characteristics are introduced in the UWB frequency range by placing a CSRR at the feeding point of the patch and another CSRR in the ground plane. Additionally, a curve shape is employed to create a notch for the satellite network band. The CSRR in the feeding line rejects the C-band, while the CSRR in the ground plane rejects the WLAN bands. Both CSRR elements increase the coupling effects and filter high-traffic bands.The 4 × 4 MIMO configuration is achieved by arranging the monopole antennas in an orthogonal configuration, providing pattern diversity and polarization diversity.High isolation within the MIMO system is achieved by utilizing the mutual coupling of CSRR elements at the patch and ground plane, effectively reducing coupling in the MIMO design system.

Interference poses challenges in wireless communication reliability. To address this issue in UWB technology, a notching approach is proposed. The UWB frequency range, designated by the FCC, spans from 3 GHz to 10 GHz. However, WLANs, WiMAX, and satellite links utilize different portions of the radio spectrum, resulting in potential interference when heavily utilized. Interactions between the UWB and occupied narrow bands lead to interference. To mitigate this interference, various parasitic slots are introduced at specific positions, and CSRR elements are incorporated into the final design to reduce interference between the UWB and narrow bands.

### 2.1. Evolution of C-Band Notch

To enhance performance and mitigate interference in the designated UWB frequency range, a notch in the C-band is introduced. By etching a square-shaped CSRR onto the patch’s feed line, the desired notching characteristic for the C-band is achieved. The CSRR length can be adjusted to fine-tune the notch. Equation (Equation 1) provides the relationship between length and frequency for the notching criterion. As the CSRR lengthens, the notch band moves down in frequency, while a shorter CSRR raises the frequency.
(1)$$S_{R}=2\left(a+b-c\right)=\frac{\lambda }{2}=\frac{c}{2f_{n}\sqrt{\epsilon_{reff}}}$$
This notching formula establishes the relationship between length and frequency for achieving the desired notch characteristics. The square-shaped Complementary Split-Ring Resonator (CSRR) on the patch feed line has a calculated wavelength of λ = 19.4 mm. By introducing this CSRR, the desired notching characteristic is achieved in the specific frequency band of 4.5 GHz as shown in Figure 2. The notching characteristics are helpful to enhance communication reliability and mitigate the interference between the UWB and narrow occupy band.

### 2.2. Evolution of WLAN Band Notch

Furthermore, when a CSRR is placed at the ground plane, it enables the attainment of a WLAN notch band characteristic. This notching characteristic effectively eliminates crosstalk between the narrow WLAN band and the UWB bands, which experience a high volume of data transfer. The corresponding wavelength of the CSRR, as determined by Equation (Equation 1), is calculated to be λ = 17.6 mm for the 5.5 GHz WLAN band shown in Figure 3.

### 2.3. Evolution of Satellite Network Notch

Satellite networks often face disruptions in communication rates due to the significant amount of data being transferred among users. By placing a C-shaped (curve-shaped) element in the patch’s center, a notching band can be created for the satellite network. The notch at the satellite network was introduced by the C-shaped curve, which also reduced interference. For the satellite network, the notch frequency was introduced at an estimated arc length of 2.8 mm as shown in Figure 4.

The notching characteristics at the C, WLAN, and satellite networks are shown in Figure 5.

The simulated and measured results are in good agreement. The simulated gain can go from −8 to 5 dB. Gain improves at in-band frequencies and degrades in notch-frequency ranges. As a result, we may say that the in-band frequency has high gain and low signal suppression, while the notched band has low gain and strong signal suppression. The gain of the antenna is represented in Figure 6. The efficiency of the specific C notch band, WLAN band, and satellite band is 48%, 52%, and 51% respectively. The efficiency is degraded in the notching band because of mismatching characteristics.

Figure 7 displays the radiation pattern’s co and cross configuration at the in-band frequency as well as the notch band. The radiation patterns at 4.5 GHz, 5.5 GHz, 8.3 GHz (notch band frequencies), and 6 GHz (in-band frequency) are shown, along with the corresponding modeled and measured radiation patterns. The results demonstrate that the proposed antenna can operate in both omnidirectional and dipole configurations.

The resonant behavior of the system can be elucidated by examining the surface current distribution on the top and bottom metallic layers, as illustrated in Figure 8. According to Faraday’s law, a time-varying magnetic field confined between two metals induces surface currents on these layers in opposing directions. As depicted in Figure 8a,b, at the resonant frequencies of 5.5 GHz and 8.8 GHz, a remarkable observation emerges: the net currents on the top and bottom layers (referred to as the ground plane) flow in diametrically opposed directions. This distinctive characteristic of oppositely directed currents on the top and bottom layers results in the attainment of a pronounced magnetic resonance at 5.5 GHz and 8.8 GHz, as evidenced in the figures. The black arrow in the diagram signifies the distribution of current on the top patch, whereas the red arrow corresponds to the current distribution on the bottom metallic layer. Furthermore, the current distribution pattern discernible in the antenna’s resonant mode bears resemblance to that of a dipole. Such behavior grants the antenna the ability to maintain stable radiation patterns across a wide frequency band, which proves to be an advantageous trait in practical applications.

The provided monopole, crescent-shaped, 4 × 4 MIMO antenna exhibits a circular configuration in Figure 1, which is advantageous for achieving pattern and polarization diversity. Antennas 1 and 2 contribute to polarization diversity, while antennas 1 and 3 enable pattern diversity. To assess the effectiveness of the monopole UWB-MIMO antenna, various critical parameters are examined. Figure 9 presents the transmission coefficient of the UWB-MIMO antenna, indicating a −25 dB transmission coefficient among the ports, which does not degrade the characteristics of the MIMO system. The high isolation in the MIMO system is achieved through the mutual coupling of CSRR.

The inter-antenna correlation is evaluated using the envelope correlation coefficient (ECC) of the UWB MIMO antenna’s diversity parameter. In a MIMO system, low correlation between antennas is desirable, and therefore the ECC (envelope correlation coefficient) value should be close to or at its minimal value, which is zero. In the case of the MIMO antenna system under investigation, it is crucial for the ECC to remain below a certain threshold throughout the entire operating frequency range. Specifically, the ECC should be less than or equal to 0.5 to ensure satisfactory performance and can be calculated using Equation (Equation 2).
(2)$$ECC=\frac{|S_{ii}^{*}S_{ij}+S_{ji}^{*}S_{jj}{|}^{2}}{\left[1-({\left|S_{ii}\right|}^{2}-{\left|S_{ji}\right|}^{2}|)\right][1-\left({\left|S_{jj}\right|}^{2}-{\left|S_{ij}\right|}^{2}|\right)]}$$

Figure 10 illustrates the ECC (envelope correlation coefficient) values, indicating the level of correlation between the antenna and the other antennas in the system. It is observed that the ECC remains consistently below 0.5 across the entire frequency spectrum, demonstrating low correlation between the antennas. In terms of diversity, the Diversity Gain (DG) should be equal to or greater than 10 (DG ≥ 10). The Diversity Gain represents the strength of the transmitted signal by the MIMO antenna system, while also indicating the reliability of the communication channel. The computation of Diversity Gain can be performed using the provided Equation (Equation 3).
(3)$$DG=10*\sqrt{1-{\left(ECC\right)}^{2}}$$

The equation establishes a relationship between the Diversity Gain (DG) and envelope correlation coefficient (ECC). When there is a higher degree of correlation among the antennas, the signal power generated by the MIMO system diminishes, leading to reduced reliability of the communication link. As depicted in Figure, the correlation between the antennas is relatively weak, and the design yields the strongest signal. The ECC value is low, while the DG value approaches 10 dB across the entire frequency range. Mean Effective Gain (MEG) represents the power received by an antenna. To ensure good diversity performance, the MEG threshold should be below MEG < 3 dB. Due to the MIMO antenna system’s excellent isolation, a low MEG value can be achieved. The provided Equation (Equation 4) can be utilized to calculate the MEG.
(4)$$MEG=\frac{1-\frac{\left(|S_{ii}{|}^{2}-|S_{ij}{|}^{2}|\right)}{2}}{1-\frac{\left(|S_{ij}{|}^{2}-|S_{jj}{|}^{2}|\right)}{2}}$$

Figure 10 demonstrates that the Mean Effective Gain (MEG) performance of the MIMO antenna remains below 3 dB throughout the entire frequency band, indicating that it does not significantly affect other antenna elements. Channel capacity loss (CCL) quantifies the amount of information lost in a MIMO antenna system. Mutual coupling between the antenna elements is the primary cause of communication loss in a MIMO setup. This mutual coupling degrades the performance of the communication system and results in losses, thereby reducing the system’s reliability. The channel capacity loss, represented by CCL, should be less than 0.4 bits per second per hertz. By analyzing the S-parameters of the MIMO antennas, the CCL can be derived using Equation (Equation 5).

Figure 10 demonstrates that the MIMO antenna system exhibits minimal mutual coupling, resulting in low channel capacity loss (CCL) throughout the entire frequency range. The low mutual coupling indicates that the communication loss in the MIMO setup is minimized, leading to enhanced system reliability. The CCL values, as depicted in Figure 10, affirm the efficient performance of the MIMO antenna system with respect to channel capacity across various frequencies.
(5)CCL=ln([AB]−[CD])
where $A=1-{\left|S_{ii}\right|}^{2}+{\left|S_{ij}\right|}^{2}$$B=1-{\left|S_{jj}\right|}^{2}+{\left|S_{ji}\right|}^{2}$

## 3. Comparison and Theoretical Analysis of the Proposed Design

Table 1 presents the theoretical comparison results of a monopole crescent-shaped antenna operating in the triple-notched frequency range. The study demonstrates that the monopole UWB antenna employs various methods to achieve notches at different frequencies. The notching in the C-band is achieved using an L-shaped stub method described in [21], while rectangular slots are used in the slot antenna for notching in WiMax and WLAN [22]. In [23], slots and PIN diodes are used for notching, and [24] employs SRR for the same purpose. Different shaped methodologies, such as W-shaped conductor and T-shaped slot, three open quarter-wavelength slots, switches, inverted V-shaped slots, embedding stubs, parasitic strips, slits, and C-shaped and parasitic stubs, are utilized to create notches at WiMax, C-band, WLAN, and the satellite network band, as observed in [25,26,27,28,29,30,31], respectively. In the proposed design, two CSRRs are employed at the feed point, ground plane, and a C-shaped curve in the center, enabling triple notching at the C-band, WLAN, and satellite networks.

Table 2 presents a comparison of different techniques for improving isolation in the MIMO antenna system. Mutual coupling, which affects the antenna characteristics and degrades the system’s performance, is a challenge in MIMO systems and can lead to spectral regrowth. To overcome this coupling effect, various techniques, such as the plus-shaped slot technique [35], decoupling structure [36], and the use of slot antenna asymmetrical placement [37], have been employed. Other techniques, including m-IFSD, common radiator, partial ground plane and slots, T slots and stub, Varactor diode, Periodic EBG, SICBS, MPS radiator, and reactive load, have also been used to mitigate the coupling effect [38,39,40,41,42,43]. In the proposed work, high isolation in a four-port MIMO antenna system is achieved by incorporating CSRR at the ground plane and patch, along with an orthogonal symmetrical distance within the system. This proposed concept minimizes the size of the MIMO antenna array while maintaining excellent isolation between the elements. The high isolation among the MIMO system can be achieved through the coupling of CSRR at the ground and patch, and then the CSRR performs multitasking in the proposed design.

## 4. Conclusions

In this article, a small-sized crescent monopole UWB MIMO antenna is presented, which demonstrates triple notching characteristics at the C-band, WLAN band, and satellite band frequencies of 4.5 GHz, 5.5 GHz, and 8.8 GHz, respectively. The implementation of Complementary Split-Ring Resonators (CSRRs) in the feeding line of the patch for C-band notching and on the ground plane for WLAN band notching, along with a curved C-shaped structure on the top of the patch for satellite network notching, successfully achieves the desired notching properties. The orthogonal positioning of the MIMO antenna system enables high isolation, pattern diversity, and polarization diversity. The diversity parameters, including ECC < 0.5, DG ≥ 10, TARC < 0, and CCl < 0.4 bits/s/Hz, are achieved, indicating improved performance. The incorporation of CSRRs in this study effectively achieves notching and high isolation for the system. The simulated and measured results are in good agreement, validating the proposed monopole crescent-shaped UWB-MIMO antenna. Overall, the presented antenna design, with its small size and desirable characteristics, is well suited for wireless communication, portable system devices, and WLAN systems.

## Figures and Tables

**Figure 1 sensors-23-06985-f001:**
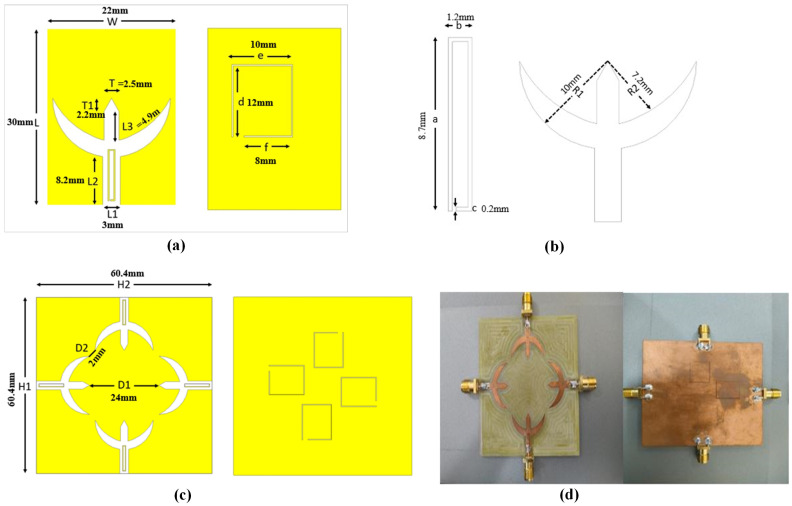
The geometrical representation of the proposed antenna: (**a**) front and back, (**b**) CSRR and arch, (**c**) MIMO antenna system, (**d**) prototype of antenna.

**Figure 2 sensors-23-06985-f002:**
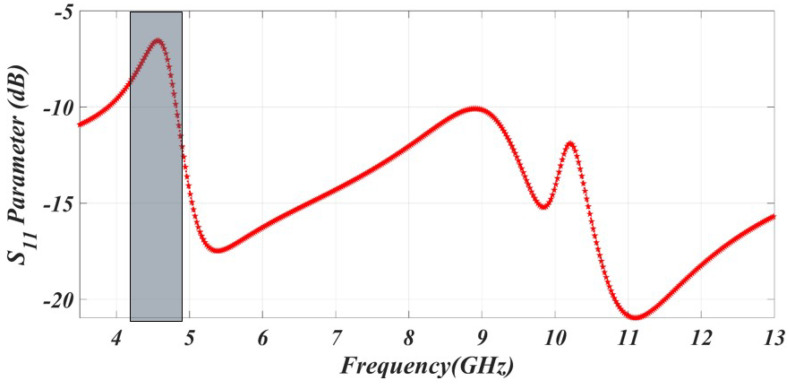
The simulated result of C-band rejection.

**Figure 3 sensors-23-06985-f003:**
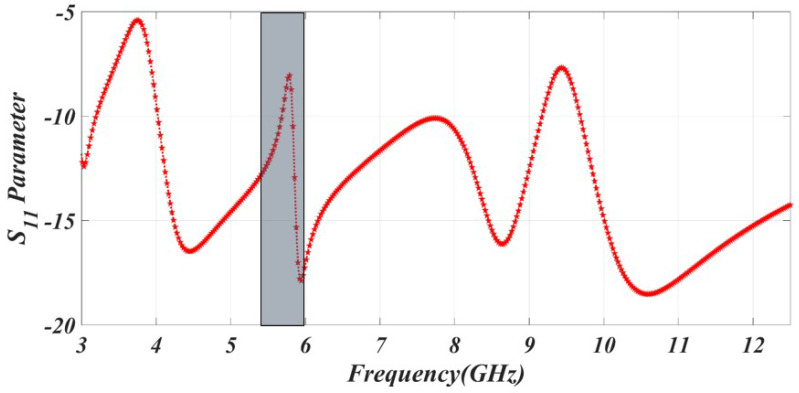
The simulated result of WLAN band rejection.

**Figure 4 sensors-23-06985-f004:**
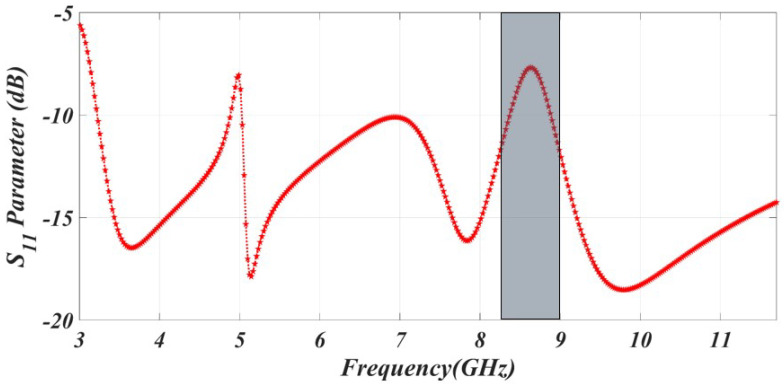
The simulated result of WLAN band rejection.

**Figure 5 sensors-23-06985-f005:**
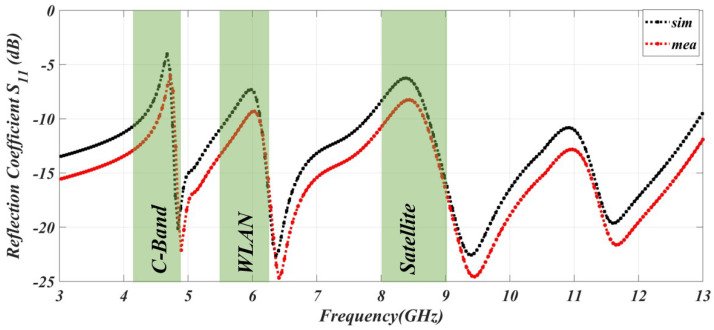
The simulated and measured result of the reflection coefficient at the C, WLAN, and satellite bands.

**Figure 6 sensors-23-06985-f006:**
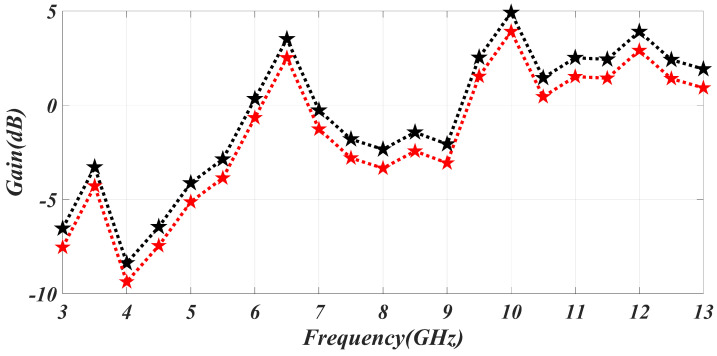
The simulated and measured result of the gain.

**Figure 7 sensors-23-06985-f007:**
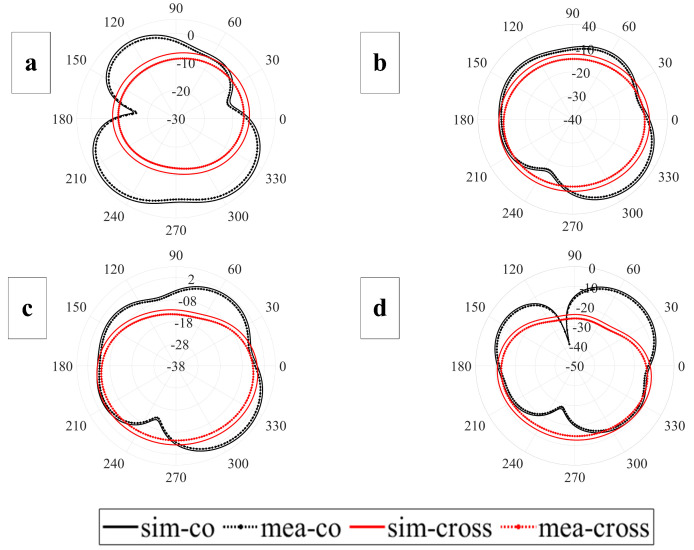
The simulated and measured results of radiation patterns at different frequencies: (**a**) C-band notch frequency 4.45 GHz, (**b**) WLAN notch frequency 5.5 GHz, (**c**) in-band frequency 6.25 GHz, (**d**) satellite notch 8.5 GHz.

**Figure 8 sensors-23-06985-f008:**
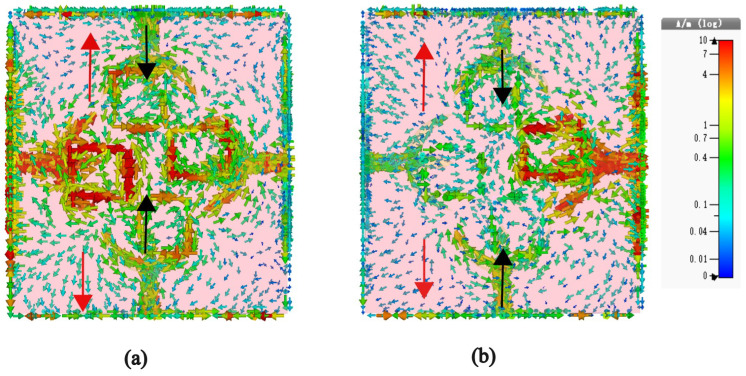
Surface current distribution at (**a**) 5.5 GHz, (**b**) 8.8 GHz.

**Figure 9 sensors-23-06985-f009:**
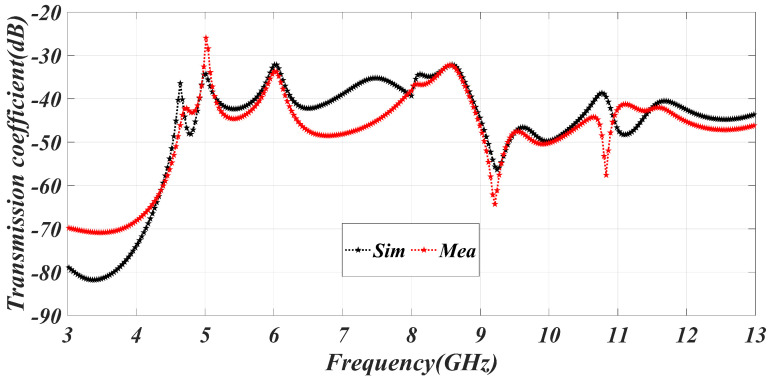
Simulated and measured transmission coefficient of MIMO system.

**Figure 10 sensors-23-06985-f010:**
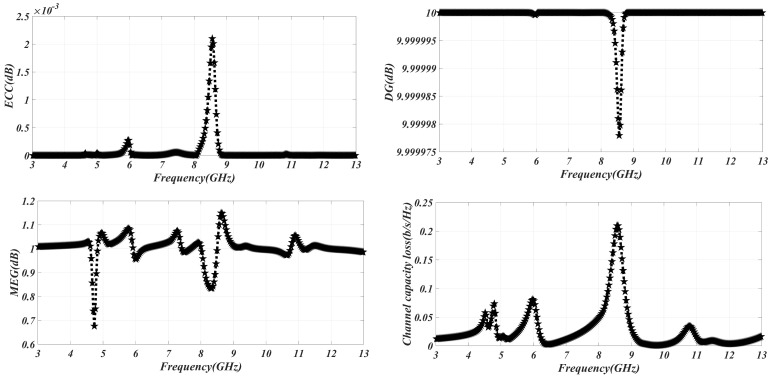
Diversity parameters ECC, DG, MEG, and CCL of MIMO system.

**Table 1 sensors-23-06985-t001:** Comparison of the proposed work’s notching characteristics with other designs.

Antenna	Technique for Notches	Notched Band	No. of Notches	Bnadwidth
MIMO or Diversity antenna [21]	L-shaped stub in the ground plane	C-band	1	3.5–4.7 GHz
Slot antenna [22]	Rectangular slits	WiMax,WLAN	2	3.3–3.7 GHz, 5.1–5.9 GHz
Slot antenna [23]	Two slots on the patch embedding PIN diodes along this slots	Wimax, WLAN	2	3.1–3.8 GHz, 5.4–6.1 GHz
Slot antenna [24]	Two Split-Ring Resonators (SRR)	WLAN	1	5.5 GHz
Monopole antenna [25]	W-shaped conductor and T-shaped slot on substrate	Wimax, C-band, WLAN	3	3.5–5.5 GHz, 5.2–5.8 GHz
Printed monopole antenna [26]	Three open-ended quarter-wavelength slots	WiMax,WLAN, X-band	3	3.3–3.7 GHz, 5.1–5.8 GHz, 7.2 GHz
Switched antenna [27]	Switches	WLAN	1	5.1–5.9 GHz
Slot antenna [28]	Inverted V-shaped slot	WLAN	1	5–6 GHz
Slot antenna [29]	Embedding two stubs	WiMax, WLAN	2	3.5 GHz, 5.8 GHz
Slot antenna [33]	Parasitic strip and U-slot	WLAN and satellite	3	5.1 GHz, 5.5 GHz, and 8.2 GHz
Circular slot antenna [34]	L and U slot	Wimax, WLAN, and satellite	3	3.2 GHz, 5.2 GHz, and 8.9 GHz
Monopole antenna [proposed work]	Two Complementary Split-Ring Resonators (CSRR) at the patch and ground	C-band, WLAN, satellite network	3	4.1–4.3 GHz, 5.2–5.8 GHz, 7.8 GHz

**Table 2 sensors-23-06985-t002:** Contrasting isolation improvement approaches with proposed work.

Reference	Dimension (mm) Material	Isolation Minimizes (dB)	Technique Employed	Application/Ports
[35]	56 × 68 × 0.2 FR-4	>−15 at 3.89–17.09 GHz	Plus-shaped slotted	UWB, X, and Ku/4
[36]	40 × 40 × 1.6 FR-4	>−20 at 3.1–11 GHz	Decoupling structure	UWB/4
[37]	45 × 25 × 1.6 FR-4	>−22 at 3.1–12 GHz	SA asymmetrical placements	UWB/4
[38]	40 × 53 RT-D5880	>−30 at 5.45 GHz	m-IFSD	-/2
[39]	130 × 90 RT Duroid	>8.6 at 1,6 GHz	Common radiator	5G, WiFi/4
[40]	120 × 60 × 1.5 RO4350	⩾−12 dB at 1.77–2.51 GHz ⩾−25 dB at 0.75–7.6 GHz	Varactor-diode-based switching techniques	UWB Cognitive Radio (CR)/five ports
[41]	90 × 45 × 1.6 FR4	⩾−30 dB at 5.5 GHz	Periodic EBG structure-based technique	WLAN/dual ports
[42]	186 × 188 × 1.6 FR-4	>30 at 2.4 GHz	Diagonal elements	WLAN/4
[43]	36 × 27 × 1.6 FR-4	−18 at 7.90–9.59 GHz	MPS radiator	X-band/4
[Proposed work]	60 × 60 × 1.6 FR-4	≥−32 dB at 3–12 GHz	Orthogonal and CSRR	UWB-MIMO/four ports

## Data Availability

Not applicable.

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
