# Peer review of "A Novel Monopole Ultra-Wide-Band Multiple-Input Multiple-Output Antenna with Triple-Notched Characteristics for Enhanced Wireless Communication and Portable Systems"

_sensors, 2023, doi:10.3390/s23156985_

Round 1
Reviewer 1 Report
Authors have reported a 4-port UWB-MIMO antenna using CSRR technique for triple notched frequency bands. The novelty in the paper is missing. I have following recommendations for this paper:
1. The abstract and conclusion sections seem similar, please rewrite the abstract portion.
2. Authors should present the design methodology in details with proper explanation.
3. In line 117 “The simulated and measured are in good comparison”, please correct the sentence.
4. Explain the antenna working using the current distribution curves.
5. In Fig 2, indicate the specific bands under consideration and with proper shading. Plot all S-parameters instead of S11, S12 and S14.
6. In Fig.2, please provide the gain and efficiency (not reported in the paper) across the applicable bands only.
7. The radiation pattern is not presented properly, radial part is missing in Fig.3. Please co and cross polar patterns at some usable frequencies.
8. In Fig 4, authors have written the Fig. caption as “Shows the diversity parameters of antenna ECC, MEG, CCL and DG”, but only two curves are visible. Please correct it.
9. In table 1 and 2 , many old papers are compared with the proposed MIMO antenna. Please include the lates papers to justify the novelty of the work.
10. Also , please include the table 1 and 2 before conclusion section.
11. In Equation 2, why S parameters are taken while calculating the ECC values of the antenna? Here, the proposed MIMO antenna is lossy, so ECC must be calculated using far fields.
12. How this paper is better than the earlier reported antennas Ref 37, 38 and 39?. The size of these antennas are much smaller than the proposed design.
minor correction
Author Response
Please find the response letter in the attached document.

Reviewer 2 Report
The authors have presented a paper titled "A Novel Monopole UWB-MIMO Antenna with Triple-Notched Characteristics for Enhanced Wireless Communication and Portable Systems". My suggestions are as follows:
1. Explain the stepwise design of the antenna
2. Add parametric study of atleast 2-3 important parameters
3. Add surface current distribution of the MIMO antenna
4. The measured value of transmission coefficient and gain should be added.
5. Kindly verify the unit of gain is dB or dBi?
6. The graph of MEG and CCL should be separated. The calculated values of ECC and DG should be compared along with simulated values
7. The Time domain analysis of the UWB antenna should be added. The authors should refer the following papers and cite them accordingly.
a. Interconnected CPW fed flexible 4-port MIMO antenna for UWB, X, and Ku band applications
b. A compact double-sided MIMO antenna with an improved isolation for UWB applications
c. A. Compact MIMO slot antenna for UWB applications
8. The dimensions of the proposed antenna should be compared in comparison table with following manuscripts
a. 4-port MIMO antenna using common radiator on a flexible substrate for sub-1 GHz sub-6 GHz 5G NR and Wi-Fi 6 applications
b. Circularly polarized 4×4 MIMO antenna for WLAN application
9. The ECC should also be calculated for realistic propagation scenarios. The authors can refer the following papers and cite them
a. “MIMO antenna mutual coupling reduction using modified inverted-fork shaped structure
b. A Dual CP Quad-Port MIMO Antenna with Reduced Mutual Coupling for X-band Application
10. References older than 2017, should be removed.
The English Language is Ok
Author Response

(The authors gave the same response as above.)

Round 2
Reviewer 1 Report
Authors have incorporated all the comments except some points which can be included in the text, however the paper is now properly organised, still some recommendations are given:
1. Please ensure the unit of the antenna gain, is it dB or dBi?
2. In Fig 9, instead of reflection coefficient, use isolation
3. In Fig 10, please check the unit of ECC
4. Figure quality should be improved
NA
Author Response
Please find our response letter in the attached document.

Reviewer 2 Report
The authors have answered all the queries hence my decision is to accept the paper. The unit of gain is dBi and not dB, author to take a note of this and do the necessary changes.
Author Response

(The authors gave the same response as above.)
